# Long Horizon Episodic Decision Making for Cognitively Inspired Robots

## Abstract

The Human decision-making process works by recollecting past sequences of observations and using them to decide the best possible action in the present. These past sequences of observations are stored in a derived form which only includes important information the brain thinks might be useful in the future, while forgetting the rest. Transformers have shown great results in multi-modal robotic navigation and human-robot collaboration tasks but lack the ability to scale to large memory sizes and learn long horizon tasks efficiently as the computational requirements needed to run these models scale non-linearly with memory length. Our model for tries to mimic the human brain and improve the memory efficiency of transformers by using a modified TransformerXL architecture which uses Automatic Chunking that chunks the past memories and only attends to the relevant chunks in the transformer block. On top of this, we use ForgetSpan which is technique to remove memories that do not contribute to learning. We also theorize the technique of Similarity based forgetting where the current observations are compared with the elements in the memory and only the new observations are stored, similar to how humans do not store repetitive memories. We test our model in various visual and audio-visual tasks that demand long horizon recollection, audio-visual instruction deciphering and robotic navigation. These tasks test the abilities of the robot that would be required in a human-robot collaboration scenario. We demonstrate that Automatic Chunking with ForgetSpan can improve the memory efficiency and help models to memorize important information and also achieve better performance than the baseline TransformerXL in the tasks previously mentioned. We also show that our model generalizes well by testing the trained models in modified versions of the tasks.

## 1 Introduction

Human cognition and decision-making works on reflection on only relevant parts of memory. We can recall specific past sequences of events in detail, without paying attention to everything in our memory. Irrelevant and repetitive parts of memory are overlooked, preferring storage of a broader picture of events based on the importance of each event. Robotic agents should have similar cognition to function well in long horizon and multi-modal tasks like navigation or human-robot collaboration. The memory buffer should be concise, containing events that will be useful for decision making in the present and future while forgetting the rest. To emulate this, in our architecture we use Automatic Chunking and ForgetSpan on the Transformer-XL memory buffer. Automatic chunking helps by chunking the memory and only using the relevant chunks in the TransformerXL layers while ForgetSpan masks out unnecessary and repetitive elements from the memory creating a more concise memory buffer which improves memory efficiency and performance. We also test a preliminary version of SimilarityWeight which decides whether a current observation should be stored in the memory by comparing it with the existing elements in the buffer.

## 2 Methodology

The main architecture of TransformerXL consists of a cyclic memory buffer which stores a specified number of pre-processed observations. The input observations are first pre-processed by a 3

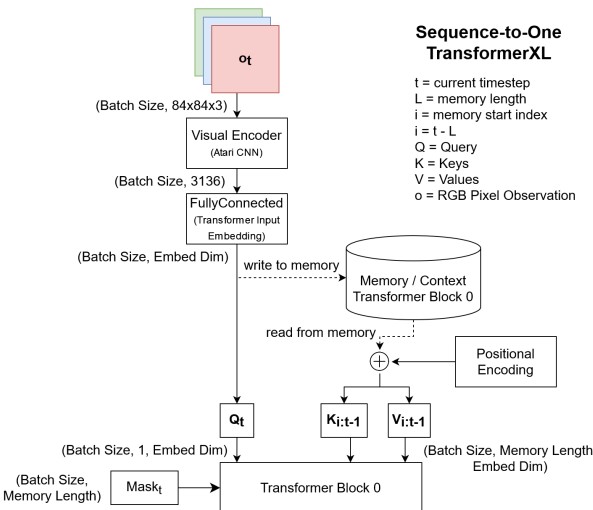

Figure 1: Transformer Mechanism Workflow

layered convolutional encoder. This encoded observation is stored in memory and also fed to the TransformerXL block as the query. The memory buffer is used to calculate the key and value in the TransformerXL block. The output of the transformer block is then used to create a categorical distribution over the action space, from which actions are sampled. PPO2 (proximal policy optimization) is used to in all models to perform consistent updates and to limit how far we can change the policy in each iteration using KL-divergence. The network policy learns to take appropriate actions based on the current observation and memory during training.

The main architecture of Automatic Chunking consists of a cyclic memory buffer which stores a specified number of processed observations and applies automatic chunking on them. The input observations are first pre-processed by a 3 layered convolutional encoder. This encoded observation is stored in memory and also fed to the TransformerXL block as the query. Memories are then split into various chunks and each chuck gets assigned a mean value which is created through mean pooling by masking. Attention is performed on these mean values to calculate the top-k chunks of memory that are relevant to the current scenario. These top-k memories are then combined and sent to the TransformerXL block to be used as the key and value in place of the whole memory buffer. The final output of the transformer block (or multiple blocks) is then used to create a categorical distribution over the action space, from which actions are samapled. PPO2 (proximal policy optimization) is used to in all models to perform consistent updates and to limit how far we can change the policy in each iteration using KL-divergence. The network learns to select relevant parts of memory and take appropriate actions based on them during training. Four tasks which test the memory, visual navigation and multi-modal instruction deciphering ability of the agent were implemented.

## 2.1 AUTOMATIC CHUNKING

In Automatic chunking, we insert our chunking algorithm in between the step where the memory buffer is passed to the TransformerXL block to calculate the key and value.The memories are instead split into sequential chunks of constant size and each chuck is assigned a mean value which is calculated through mean pooling by masking. Attention is performed on these mean values to calculate the top-k chunks of memory that are relevant to the current scenario. These top-k memories are then combined and sent to the TransformerXL block to be used as the key and value in place of the whole memory buffer. This reduces the number of memories that need to be attended by the transformer block as well as provides more contextual information to it. The chunk size and number of chunks used in our experiments have been detailed in the appendix.

## 2.2 FORGETSPAN

In ForgetSpan, we use masking to remove memories from the memory buffer after a certain span of time. The model learns to decide the span each memory element will stay in the buffer using the logic below and removes memories that do not contribute to learning. This allows the transformer to learn from a more concise memory buffer, improving learning as well as reducing the memory requirement of the model.

We calculate a ForgetSpan $f_i \in [0, L]$ for every element in memory $m_i$

$$f_i = F\sigma(W^T m_i + B) \tag{1}$$

Here $W$ and $B$ are a trainable weight and bias, $sigma$ is a sigmoid function for activation and $F$ is the maximum span an element can stay in memory. This allows us to determine a singular value of ForgetSpan for every element in the memory, which the model learns to calculate more efficiently as it trains. We calculate the remaining span $r_{ti}$ at every timestep $t$ for the $i^{th}$ memory element.

$$r_{ti} = f_i - (t - i) \tag{2}$$

When $r_{ti}$ becomes negative, it means the element has to be forgotten and can be masked out of the memory buffer. We use a soft masking function that creates a smooth mask from $1$ to $0$ once the element has to be forgotten.

$$s_{ti} = max(0, min(1, 1 + r_{ti}/R)) \tag{3}$$

Where $R$ is the ramp length of the ramp between $1$ and $0$. This allows $f_i$ to receive a gradient to train as the masking function has a non-zero gradient between $[-R, 0]$. The parameters for ForgetSpan used in our experiments are detailed in the appendix.

## 2.3 SIMILARITYWEIGHT

In SimilarityWeight we calculate the similarity between the current observation with all the elements currently in the memory buffer using cosine similarity. We then bin the similarity values into 10 bins and calculate the number of values in the top k bins. We use k=3 for our experiment in the Minigrid Task. This number is used to represent the similarity of the current element with the memory as it denotes that the number of memories the current observation is highly similar with. A dynamic threshold is used which updates every n timesteps and is used to decide whether the current observation is stored in memory or not. If the value of similarity is greater than the threshold, that means the memory is highly similar to the memory buffer and so it is not stored. If the value of similarity is lower than the threshold, the memory is stored. Where $similarity$ is:

$$similarity = topkbins(cossimi(obs, memory)) \tag{4}$$

Where topkbins is the function to bin and choose the top-k highest populated bins. Cosine similarity is calculated using the torch.nn.CosineSimilarity function. SimilarityWeight is employed to remove new observations that are extremely similar to elements already in the memory thus creating a small memory with highly focused elements.

## 2.4 TASKS

The Minigrid memory task was used to test the models for 2D tasks. Unity MLAgents toolkit was used for implementing the Audio-Visual Instructions Task and Visual Corridor task. These tasks are chosen as they deal with the memory, navigation, locomotion and multi-modal aspects of robotic agents that will be required in real-world applications. Various combinations of Gated TransformerXL, Automatic Chunking, ForgetSpan and SimilarityWeight were tested in all the tasks to see the effects on training performance.

## 3 RESULTS AND DISCUSSIONS

### 3.1 MINIGRID MEMORY TASK

The goal of this task is to correctly remember the object seen in the initial room (on the left) and then navigate to the end of the corridor and touch the same object. The agent's observation space includes

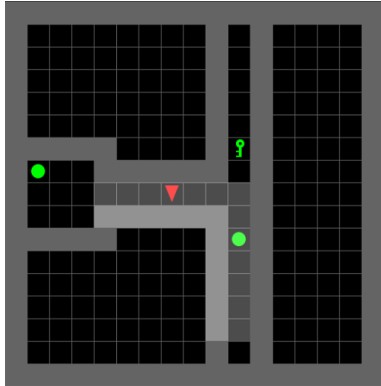

Figure 2: Minigrid Task

a 5x5 square image of the grid ahead of the agent. The action space is discrete with the actions Turn left, Turn right and Move forward. This task tests the agent's ability to remember the information at the start of the episode and use it effectively to reach the final goal. Automatic chunking and ForgetSpan are 0

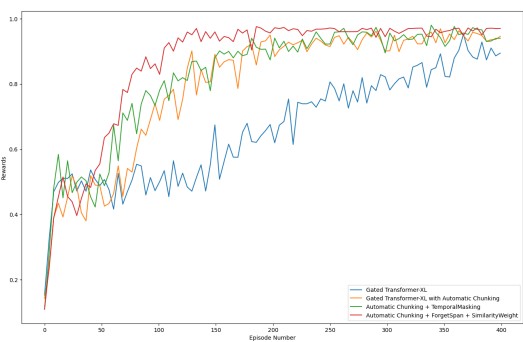

Figure 3: Minigrid Task training rewards

In Fig 3, we plot then average rewards across episodes for Baseline Gated TransformerXL, Gated TransformerXL with Automatic chunking, Gated TransformerXL with Automatic chunking and ForgetSpan and Automatic chunking with ForgetSpan and Similarity Weight. Automatic Chunking with ForgetSpan learns the task slightly faster than Gated TransformerXL with Automatic chunking which in turn trains faster than the baseline Gated TransformerXL. Automatic Chunking with ForgetSpan and SimilarityWeight gives the best results by training the fastest and with the highest reward. However, the average training time required for each episode is 17 sec, 46 sec, 20 sec and 42 sec for Gated TransformerXL, Automatic Chunking with TransformerXL, Automatic Chunking with ForgetSpan and Automatic Chunking with ForgetSpan and SimilarityWeight respectively. The computational cost was reduced greatly by ForgetSpan while maintaining the performance of Automatic Chunking. Automatic Chunking with ForgetSpan and SimilarityWeight gives the best results by pre-processing the memories and complimenting Automatic Chunking but the computational cost increase is significant as we have to calculate similarity of a new observation with the whole memory buffer every timestep.

## 3.2 AUDIO VISUAL INSTRUCTIONS TASK

In this task the agent gets one of two audio commands randomly at the start of each episode, either "red cube" or "green cube". The agent then has to navigate based on visual inputs to the specified cube. The observation space consists of audio spectrograms of size 41 X 42 X 1 along with visual observations of size 41 X 42 X 3. This task tests the agent's recollection as well as multi-modal instruction deciphering ability. The episode ends whenever the agent touches one of the objects.

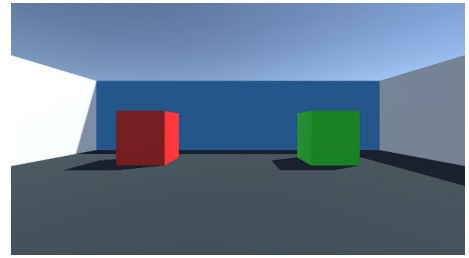

Figure 4: Audio-Visual Instructions Task

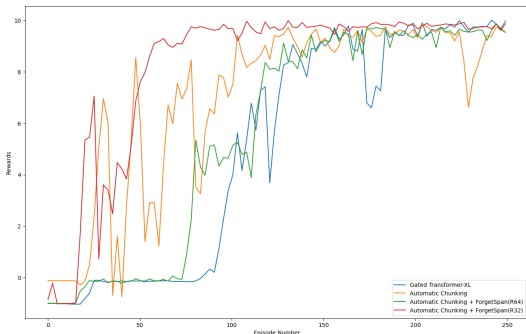

Figure 5: Audio-Visual Instructions Task Rewards

In figure 5, we plot the average training rewards for Gated TransformerXL, Gated TransformerXL with Automatic chunking, Automatic chunking TransformerXL with ForgetSpan with ramp length 64 and 32. As we can see Automatic chunking with ForgetSpan with Ramp length 32 has the best performance with the highest rewards and fastest training. Increasing the Ramp length to 64 led to worsening performance. This is probably caused by the gradient used to train the ForgetSpan is more gradual leading to slower learning of the ForgetSpan layer. Automatic Chunking TransformerXL performed better than baseline TransformerXL while being better than ForgetSpan with ramp length 64 and worse than ForgetSpan with ramp length 32. To test whether our model is generalizable we tested the trained model on two scenarios, one with the boxes in the same positions as in training and the other where their positions were changed. This would test if the model had learned a proper mapping between audio and visual observations. Figure 6 shows the rewards gained by the model in 30 test episodes. As we can see the model was able to go to the correct target most of the time, proving that it had learnt a proper mapping. The episodes where it missed the targets could be attributed to the agent travelling past the boxes.

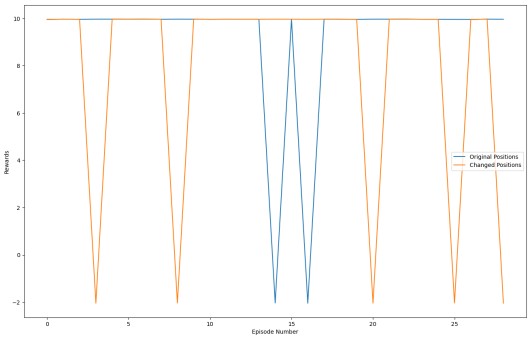

Figure 6: Audio-Visual Instructions Task Testing Rewards

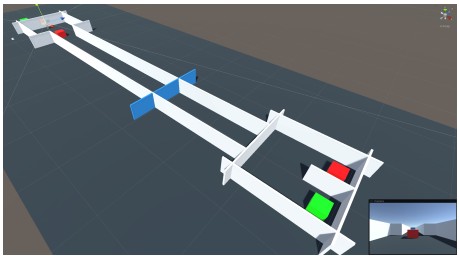

Figure 7: Visual Corridor Task environment in Unity

### 3.3 VISUAL CORRIDOR TASK WITH VARIABLE DISTRACTOR

In this task the agent observes one of the two cubes either red or green in colour at the start of each episode. The agent then has to navigate along a long corridor of variable length until it reaches the end at which time it is teleported to the final room where it has to go to the cube it saw at the start of the episode. The observation space consists of visual observations of size 40 X 40 X 3 and the position of the agent. This task tests the agents ability to recall information after a variable distractor phase. We only tested Automatic Chunking with ForgetSpan in this task as we wanted to test the forgetting of ForgetSpan as well as the chunk selection of Automatic Chunking in a more dynamic scenario.

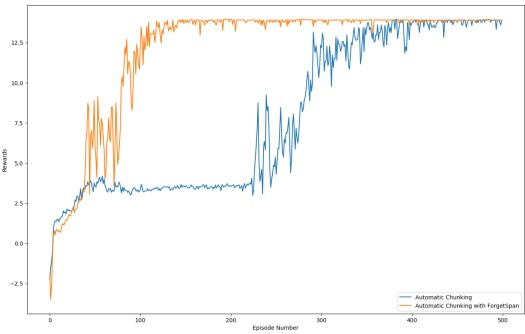

Figure 8: Visual Corridor Task Rewards

In figure 8 we can see that Automatic Chunking with ForgetSpan using ramp length 100 trained by 150 episodes and learnt to do the task even with the variable distractor phase. Automatic chunking without ForgetSpan took longer to train but reached the same final rewards. This shows that ForgetSpan improves training performance of Automatic Chunking significantly while also improving memory efficiency. To test whether our model was generalizable, during the test scenario we doubled the length of the variable distractor and tested for 30 episodes. Both models managed to reach the final goal for 30 out of 30 episodes as shown by the approximately 15 reward received by each of them every episode in figure 9.

## 4 CONCLUSION

Transformers with Automatic chunking and memory handling techniques like ForgetSpan and SimilarityWeight showed better memory efficiency and performance over regular transformers models in memory, robot navigation, and multi-modal tasks. Automatic chunking improved the baseline TransformerXL by giving a more focused memory for the transformer block to attend to. ForgetSpan and SimilarityWeight showed good synergy with Automatic chunking, improving the training speed as well as the memory efficiency of the model by creating a concise memory with only relevant memories for the transformer architecture to work on. This work aims to improve the performance of Robotic agents in Human-Robot Collaboration tasks which are generally multi-modal, long horizon and dynamic in nature and would greatly benefit from human-like memory. Automatic

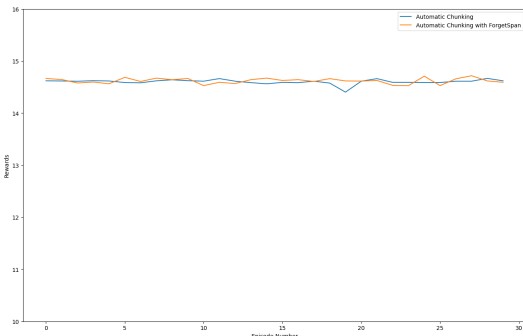

Figure 9: Visual Corridor Test Rewards

Chunking, ForgetSpan and SimilarityWeight are a step towards emulating human-like cognition in robots.

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

## A   APPENDIX

All the hyperparameters used while training the models in this research work are listed below.

### A.1   PPO PARAMETERS

- learning rate(initial): 3e-4 (decays consistently during training, final value is 3e-5)
- gamma: 0.995
- lambda: 0.95
- updates: 100
- epochs: 5
- n workers: 20
- n mini batch: 10

The above standard Proximal policy optimization parameters were chosen with extensive testing for the purpose of making sure that optimal behaviour is learnt within 100-150 training episodes for the standard transformer-XL model. These parameters were kept the same across all the models used in this research work in order to obtain appropriate comparative results. All tasks made use of 20 workers and 10 minibatches in order to reduce training time.

### A.2   TRANSFORMER PARAMETERS

- embed dim: 250
- number of heads: 5
- memory length: 64
- positional encoding: True
- gating: True

With extensive testing, the above parameters were changed based on the task in order to speed up training and get stable results. However, the same values were taken during comparative study with different architectures. The embed dimension parameter specifies the common dimension to which the keys, queries and values will be converted to make the multilevel attention mechanism work. The number of heads parameter specifies the amount of transformer heads. For all tasks, the embed dimension was 250 and the number of heads were 5. Both positional encoding and layer normalization were set to true for all the tasks to ensure that proper and effective sequence processing is performed by the transformer. The memory length parameter specifies the amount of time step information stored in the memory buffer. Memory length for the Minigrid task was set to 320 and for the audio-visual navigation task it was set to 384. For the visual corridor task, it was set to 500. The Gating parameter was used to decide whether a gating mechanism is implemented.

### A.3   MEMORY PARAMETERS

- n chunks: 3
- chunk size: 50
- max span: 250
- ramp length: 50

The chunk size and number of chunks denote the length and number of sequential events being selected during training. The number of chunks were set to 3 for all experiments. For minigrid and audio-visual instructions task , the chunk size was set to 80. For the visual corridor task the chunk size was set to 100. In the minigrid task, the max span of ForgetSpan was kept to be the memory size of 320 with the ramp length being 50. For the audio-visual instructions task, the max span was set to the memory size of 384 and the the ramp length was 32. For visual corridor task, the max span was set to the memory length of 500 and the rap length was 100

