# OpenReview forum: "Long Horizon Episodic Decision Making for Cognitively Inspired Robots"
_ICLR.cc/2024/Conference — Submitted to ICLR 2024_

### Official Review · Reviewer_nji3 · 2023-10-20

**Soundness:** 1 poor
**Presentation:** 1 poor
**Contribution:** 1 poor
**Rating:** 3
**Confidence:** 5

**Summary:**

This work approaches the problem of dealing with long horizon tasks in decision making systems (such as Robotics), particularly with the transformer architecture, whose memory costs grow non-linearly with the memory length. For this matter, it introduces three techniques to be applied on top of the TransformerXL architecture: Automatic Chunking, Forget Span, and SimilarityWeight. These techniques implement heuristics for retrieval and forgetting of non-parametric memories based on the similarity of state representations. The work presents experiments on a set of memory tasks, ranging from gridworld to multi-modal navigation tasks. It claims improvements in sample efficiency by using the proposed techniques.

**Strengths:**

- The work brings an interesting set of tasks that focus on memory aspects and progressively increases the difficulty on the representation and environment dynamics, which is ideal to evaluate the claims in the work.

**Weaknesses:**

- The work has a major flaw in terms of clarity. First, the structure can be greatly improved. In more details:
     - The Introduction could be more clear on the contributions of the paper and better describe the motivations;
    - The Section 2 could be better modularized in two:
        - A Preliminaries Section, to introduce, for instance, the problem formulation, the RL algorithm, and the transformerXL architecture);
         - A proper Methodology section (describing what the work is proposing);
    - Section 3 could also be better structured, by first bringing the experimental setup (with task description, training setup) and then raising results and discussions later on.
    - During most of the work, the sentences are very long and the reasoning is confusing; also, the description of the techniques is repeated several times in different parts of the work (for instance, the Automatic Chunking technique is described during Section 1 and 2, and Subsection 2.1).
    - The mathematical formulas should also be rewritten for better clarity. The paper should define mathematical symbols and use them to present equations.
    - Some part of Section 3.1 (above Figure 3) is simply cut off. There are also typos and missing full stops. Crucially, the paper must be proofread carefully before an official submission.


- In the Abstract and Introduction, the work starts with strong claims regarding the human cognition, and decision-making process, to serve as motivation for the work. Nevertheless, these claims are not backed-up with proper literature and state general assumptions on how the memory works, which is actually an active topic of research in many disciplines.

- In the same line, the work does not bring any literature review. The study of memory systems in RL is very active nowadays, as well as transformers for RL. Therefore, the work must discuss related work and contrast with prior similar techniques, to contextualize the readers and clarify the contributions of what is being proposed.
    - In the same topic, the Reference section does bring some works, which are not properly cited in the text. This suggests that something is wrong with the template files used to write this paper.

- There are many design choices that are not clear or unjustified. I will raise them as questions below. Furthermore, the choice of a transformer architecture is not motivated. Transformer optimization is often hard, especially in RL setting [1], so why is that required for the proposed formulation?

- Without a proper argument in the paper to contrast prior work, it seems to lack originality in the proposed techniques. Indeed, there are several works on non-parametric memory with forgetting heuristics ([2, 3] to name a few), which should be contrasted and compared against in order to evaluate the effectiveness.

- There are also flaws in the experimental execution:
    - No experiment brings confidence intervals, which raise questions about statistical significance. This is crucial on RL experiments given the intrinsic variance of methods and non-determinism of benchmarks [4, 5].
    - The work does bring some ablation on top of the transformerXL architecture, but there are no baselines on memory architectures to compare against. For instance, the paper could reproduce the methods in [2, 3] to serve as baselines.
    - In Figure 8, there is no baseline at all, which makes it hard to evaluate the effectiveness of the proposed method.
    - In Appendix A, some information seems confusing and even contradictory. For instance, please look at the chunk size information in Appendix A.3 in the list and text description.

**Further Suggestions/Minor Concerns:**

- It would be interesting to bring a Pseudocode for the proposed techniques. This would help to formalize the proposed heuristics and better situate the reader in the algorithmic flow.

- As a way to improve clarity, the environment descriptions could be moved to an appendix, in order to decouple from the experiment's results and discussion. Otherwise, it breaks the reading flow by switching the context from setup description to analysis.

- Another way to improve clarity is to present some illustrations of the proposed techniques. Figure 1 is confusing, as it is hard to know what is part of the transformerXL and what is the contribution.  A schematic illustration, clearly identifying which of the proposed modules, would be very helpful.


**References**

[1] Parisotto et. al. Stabilizing Transformers for Reinforcement Learning. ICML, 2020.

[2] Yalnizyan-Carson and Richards. Forgetting Enhances Episodic Control With Structured Memories. Frontiers of Computer Neuroscience, 2022.

[3] Coda-Forno et. al. Leveraging Episodic Memory to Improve World Models for Reinforcement Learning. Memory in Artificial and Real Intelligence at NeurIPS, 2022.

[4] Henderson et.al. Deep Reinforcement Learning that Matters. AAAI, 2018.

[5] Agarwal et.al. Deep Reinforcement Learning at the Edge of the Statistical Precipice. NeurIPS, 2021.

**Summary of the Review**

The work brings several major flaws and needs a careful revision with rewriting in order to clearly state its claims, motivations, methodology, and experimental results.

**Questions:**

- What is PPO2 and how is it different from PPO? What is the reason behind this “2”?

- In Section 2.2, the elements to be forgotten are masked out. Why not delete them entirely? This would cost less memory in the long run.

- In Section 2.3, the paper states that the dynamic threshold is used with updates every n timesteps, but what is the value of n in the experiments? Also, how are the bins constructed? Answering these questions in the paper is vital for the sake of reproducibility.

---

### Official Review · Reviewer_KtCs · 2023-11-01

**Soundness:** 2 fair
**Presentation:** 1 poor
**Contribution:** 3 good
**Rating:** 1
**Confidence:** 4

**Summary:**

The paper discusses the use of memory mechanisms in language processing and reinforcement learning agents. It introduces the concept of Automatic Chunking, a model that combines the TransformerXL architecture with a cyclic memory buffer and chunk selection mechanism. The model is tested on two tasks: an audio-visual instructions task and a visual corridor task with a variable distractor. The results show that Automatic Chunking with ForgetSpan improves training performance and memory efficiency. The model demonstrates the ability to learn a proper mapping between audio and visual observations and recall information after a variable distractor phase.

**Strengths:**

This paper offers a pioneering approach to enhancing memory efficiency in transformers by emulating human cognitive processes. Its introduction of a modified TransformerXL architecture is both innovative and timely. The paper further stands out with its trio of novel techniques: Automatic Chunking, ForgetSpan, and Similarity based forgetting, each addressing unique memory optimization challenges. Comprehensive testing across diverse tasks not only validates the model's robustness but also underscores its relevance in real-world robotic applications. In essence, the paper shines in its blend of innovation, rigorous evaluation, and practical applicability.

**Weaknesses:**

- Some statements require more evidence and clarification.
  - Introduction: "Human cognition and decision-making works on reflection on only relevant parts of memory.". Should add citations of related works to support argument.
  - Introduction: "Robotic agents should have similar cognition to function well in long horizon and multi-modal tasks like navigation or human-robot collaboration.". Should add citations of related works or analysis to support argument.
- Lack of comparative analysis
  - The paper primarily compares the proposed model with the baseline TransformerXL. A more extensive comparison with other state-of-the-art models might provide a clearer picture of the model's standing in the current research landscape.
- Limited discussion about result
  - Most discussions only state the data provided in the figure and concluded as "our method is better than baseline", and further discussion about the result is missing.

**Questions:**

- How do you tackle the overfitting issue? While the paper mentions that the model generalizes well by testing on modified versions of tasks, there might be concerns about overfitting, especially if the modifications are not significantly different from the original tasks.

---

### Official Review · Reviewer_giAH · 2023-11-02

**Soundness:** 1 poor
**Presentation:** 1 poor
**Contribution:** 1 poor
**Rating:** 1
**Confidence:** 5

**Summary:**

This paper attempts to introduce a method for long-horizon episodic decision-making for cognitively inspired robots.

**Strengths:**

+ The problem of long-horizon episodic decision-making seems important for cognitively inspired robots, although not well justified in the paper.

**Weaknesses:**

- The paper's novelty is considerably low.

- The paper does not show an understanding of the state of the art.

- Experimental results cannot justify the novelty or contribution of the proposed approach.

- The paper is poorly written with many grammar errors and typos.

**Questions:**

Please see the weaknesses section.

---

### Official Review · Reviewer_Kvk7 · 2023-11-03

**Soundness:** 1 poor
**Presentation:** 1 poor
**Contribution:** 2 fair
**Rating:** 1
**Confidence:** 4

**Summary:**

This paper proposes a novel transformer architecture TransformerXL with improved memory efficiency, achieved through selective encoding of memories that contribute to learning, and discarding memories determined to be not relevant. This project seems to be in an early stage, there is no related work section, and methodology section does not explain the model well enough for someone to try to replicate it.

**Strengths:**

I think the memory implementation is original, but because of poor presentation it is hard to understand the significance of the contribution.

**Weaknesses:**

This paper is not ready yet for a conference presentation, it is an early stage technical report.

**Questions:**

Model is compared to its own ablated versions, but not to existing models - why is that?

---

### Meta-Review · Area_Chair_LeWb · 2023-12-05

**Metareview:**

Summary: This paper focuses on the use of memory mechanisms in language processing and reinforcement learning agents. Specifically, it introduces Automatic Chunking, which combines a transformer architecture with a cyclic memory buffer and chunk selection mechanism. Reviewers agree that the project is in early stages, lacking related work or a strong understanding of the SotA, with unclear writing and grammatical errors, and not enough experimental evidence and comparisons to SotA.

Strengths:
- The modified TransformerXL architecture is interesting.
- The chosen evaluation tasks are interesting and well suited.

Weaknesses:
- There's little to no comparison to SotA methods (most experiments are just ablations).
- The paper does not present enough related work, which puts into question the understanding of SotA methods.
- The writing in the paper is unclear and rife with typos and grammatical errors.
- There are not enough methodology details to allow replication. There are also many design choices that are not clear or unjustified.

**Justification For Why Not Higher Score:**

All reviewers agree that this paper lacks proper experimental baselines and comparisons, a thorough understanding of the SotA in the field, and writing clarity. I agree that this paper is in very early stages and does not yet merit publication.

**Justification For Why Not Lower Score:**

N/A

---

### Decision · Program_Chairs · 2024-01-16

Reject